# Genome-Wide Identification and Expression Analysis of the *SWEET* Gene Family in Annual Alfalfa (*Medicago polymorpha*)

**DOI:** 10.3390/plants12101948

**Published:** 2023-05-10

**Authors:** Nana Liu, Zhenwu Wei, Xueyang Min, Linghua Yang, Youxin Zhang, Jiaqing Li, Yuwei Yang

**Affiliations:** 1College of Animal Science and Technology, Yangzhou University, Yangzhou 225009, China; 2Institute of Grassland Science, Yangzhou University, Yangzhou 225009, China

**Keywords:** *Medicago polymorpha*, SWEET, phylogenetic evolution, expression analysis, abiotic stress

## Abstract

SWEET (Sugars will eventually be exported transporter) proteins are a group of sugar transporters that are involved in sugar efflux, phloem loading, reproductive development, plant senescence, and stress responses. In this study, 23 SWEET transporter members were identified in the *Medicago polymorpha* genome, heterogeneously distributed on seven chromosomes. These *MpSWEET* genes were divided into four subfamilies, which showed similar gene structure and motif composition within the same subfamily. Seventeen *MpSWEET* genes encode seven transmembrane helices (TMHs), and all MpSWEET proteins possess conserved membrane domains and putative serine phosphorylation sites. Four and three pairs of *MpSWEET* genes were predicted to be segmentally and tandemly duplicated, respectively, which may have contributed to their evolution of *M. polymorpha*. The results of microarray and RNA-Seq data showed that some *MpSWEET* genes were specifically expressed in disparate developmental stages (including seedling stage, early flowering stage, and late flowering stage) or tissues such as flower and large pod. Based on protein network interaction and expression patterns of *MpSWEET* genes, six *MpSWEET* genes were selected for further quantitative real-time PCR validation in different stress treatments. qRT-PCR results showed that *MpSWEET05*, *MpSWEET07*, *MpSWEET12*, *MpSWEET15*, and *MpSWEET21* were significantly upregulated for at least two of the three abiotic stress treatments. These findings provide new insights into the complex transcriptional regulation of *MpSWEET* genes, which facilitates future research to elucidate the function of *MpSWEET* genes in *M. polymorpha* and other legume crops.

## 1. Introduction

Soluble sugars are the major source of carbon skeletons and energy for living organisms. As the main components of plant cell structures, carbohydrates (including glucose, sucrose, and fructose) play key roles in storing energy and regulating cell metabolism [1]. Sugars are synthesized in source organs (carbon sources) and then translocated sink organs (carbon sinks). Importantly, all sink organs receive an adequate supply of sugars for growth and development [2], and sugar transporters play a crucial role in the membrane transport of sugars and their distribution throughout the plant [3,4]. In plants, sugar transporters are divided into three categories: monosaccharide transporters (MSTs), sucrose transporters (SUTs) [5,6], and SWEET (Sugar Will Eventually be Exported Transporters) families: mediating influx or efflux of sugars from phloem parenchyma into the phloem apoplast [7,8].

The SWEETs are a novel family of sugar transporters that have been discovered in recent years, belonging to the MtN3-like clan, with a typical MtN3/saliva domain [9,10], which is essential for the maintenance of animal blood glucose levels [10], plant nectar secretion [11], and pollen nutrition [12]. In eukaryotes, SWEET proteins contain seven transmembrane helices (TMHs), consisting of three TMH units of two tandem repeats, separated by a single TMH [9,13]. Phylogenetic analysis has shown that SWEET proteins can be divided into four clades [10], with clades I and II preferentially transporting hexoses [9,14], clade III transporting sucrose [15], and clade IV transporting fructose [16]. Up until now, SWEET transporter members have been identified in many plant species, including *Arabidopsis* [10], rice [17], wheat [18], *Medicago truncatula* [19], tomato [20], cucumber [21], cotton [22], strawberry [23], and walnut [24]. In legume crops, the highest number of 52 SWEETs was identified in soybean, while only 16 SWEETs were identified in *Lotus japonicas* [25].

Studies have shown that SWEETs influence a wide range of physiologically important processes. For example, AtSWEET11 and 12 sucrose efflux transporters are responsible for the first step in phloem loading of sucrose for long-distance sugar transport within plants [15]. *OsSWEET11* encodes a rice sucrose uniporter that is specifically expressed in phloem cells, indicating that it may participate in phloem sucrose loading in rice [26]. Besides, some SWEETs regulate plants’ reproductive development such as pollen, nutrition, and seed filling. *AtSWEET8* participates in the development of the pollen and anther—the suppression of which was shown to reduce starch content in pollen grains and cause male sterility [27]. *AtSWEET11*, *12*, and *15* exhibit specific spatiotemporal expression patterns in developing *Arabidopsis* seeds, in the *atsweet11*, *12*, and *15* triple mutant, embryo development, seed weight, and composition are severely affected [28]. Soybean *GmSWEET10a*/*b* mediates the transport of soluble sugars from seed coat to embryo, which can provide a carbon source for seed development and oil accumulation [29]. *LcSWEET2a* and *LcSWEET3b* genes that regulate early seed development were also preliminarily identified in litchi [30]. In addition, SWEET transporters are also involved in host–pathogen interaction during plant growth and development and responses to abiotic stresses by maintaining intracellular sugar concentrations. For example, in *Vitis vinifera*, *VvSWEET4* expression was reported to participate in the interaction with *Botrytis cinerea* [31]. *MtSWEET1b* is specifically localized to the peri-arbuscular membrane. Overexpression of *MtSWEET1b* in *M. truncatula* roots promoted the growth of intraradical mycelium during arbuscular mycorrhizal (AM) symbiosis, revealing SWEET transporter may provide sugar to AM fungi to maintain a successful symbiosis [32]. *AtSWEET4* expression was reported to enhance tolerance to freezing and drought stresses [33]. *atsweet11*/*12* double mutant exhibited higher freezing tolerance due to the high accumulation of sugars in the leaves [34]. *SAG29*/*AtSWEET15* was found to be associated with cell viability under high salinity and other osmotic stress conditions [35]. *AtSWEET16* and *AtSWEET17* are involved in the transport of monosaccharides and polysaccharides across tonoplast [36]. Overexpressing *AtSWEET16* increased the tolerance to freezing stress and improved germination as well as nitrogen use efficiency in *Arabidopsis* [36]. Similarly, the cold-suppression gene was identified in tea, and *CsSWEET16* contributed to sugar compartmentation across the vacuole and function in modifying cold tolerance in *Arabidopsis* [37].

*M. Polymorpha*, which is widespread around the world, is recognized as a nutritious and palatable forage plant. Being annual alfalfa, *M. polymorpha* is closely related to the evolution of *M. truncatula* and *Medicago sativa*, but the lignin content of *M. polymorpha* was lower than *M. sativa* during the same growth period, and it had higher crude protein content. The crude protein percentages of different cultivars ranged between 17.8% and 22.2% and showed good digestibility for use as forage [38,39]. *M. polymorpha* also has the ecological value of efficient nitrogen fixation and soil improvement [40] and is one of the traditional green manure crops in southern China. Moreover, *M. polymorpha* is rich in nutrients and has edibility, and is consumed both cooked and fresh in China. However, to our knowledge, no systematic investigations on the *SWEET* gene family of *M. polymorpha* have been reported to date. In this study, we present a detailed analysis of the *M. polymorpha SWEET* family members, including the phylogeny, conserved domain, gene structure, chromosome distribution, *cis*-acting regulatory elements, and expression patterns in different developmental stages/tissues, as well as in response to drought, salt, and cold stresses. The results will provide a solid foundation for further characterizing the functions of SWEET proteins in the regulation of *Medicago* plants’ development and stress responses.

## 2. Results

### 2.1. Identification and Phylogenetic Analysis of MpSWEET Genes

A total of 23 SWEETs (MpSWEETs) were identified in the *M. polymorpha* genome, and these proteins were named MpSWEET1-MpSWEET23 in order, according to their chromosomal position. The physical and chemical characteristics of the MpSWEET proteins were predicted and are listed in Table 1. Protein length sizes were found to range from 211 aa (MpSWEET16) to 305 aa (MpSWEET12). The predicted molecular weights range from 24.51 kDa (MpSWEET01) to 34.56 kDa (MpSWEET12), and the theoretical isoelectric points (pI) range from 5.13 (MpSWEET16) to 10.21 (MpSWEET03), with all, except for three, being higher than 7.60. The GRAVY (Grand average of hydropathicity) of MpSWEET proteins is larger than 0, indicating that all MpSWEETs were hydrophobic proteins. The Aliphatic Index range from 106.92 (MpSWEET12) to 129.33 (MpSWEET23), revealing that MpSWEET protein is lipolysis. The prediction of the subcellular localization of the MpSWEETs revealed that most of the proteins are localized to the plasma membrane or chloroplast (Table 1).

To explore the evolutionary relationships between MpSWEETs, the protein sequences of 86 SWEETs from *M. polymorpha*, *Arabidopsis* (AtSWEETs), *M. truncatula* (MtSWEETs), and rice (OsSWEETs) were aligned for the construction of an unrooted phylogenetic tree, and these members were subdivided into four clades, containing 22, 27, 31, and 5 members, respectively. As shown in Figure 1, MpSWEETs also clustered into four clades. Clade III contains the most MpSWEET proteins, with 9 members, clade IV contains the fewest members, containing only one MpSWEET21, and clades I and II contain 6 and 7 MpSWEET proteins, respectively.

### 2.2. ProteinConserved Domains Analysis

The predicted results of TMHMM server 2.0 suggested that seventeen MpSWEET proteins contained seven TMHs, five proteins had six TMHs, and one had five TMHs (Table 1, Appendix A). In soybean, there are also some GmSWEET proteins that have five and six TMHs, not seven TMHs [25].

Multiple sequence alignments of the deduced protein sequences performed were used to obtain more detailed information concerning the MpSWEET proteins and analyze the conserved amino acid residues. As shown in Figure 2, the residues represented by a black background are completely conserved in all proteins, and it is speculated that these amino acid residues may play an important role in the function of the SWEET protein [41].

All MpSWEET proteins retain relatively conserved membrane domains and contain the active sites of tyrosine (Y) and aspartic acid (D), which can form a hydrogen bond to maintain the sugar transport activity [41]. The obvious feature of eukaryotic SWEETs is their long cytosolic C-terminus, which carries multiple phosphorylation sites [13]. Each MtN3/slv domain contain a conserved serine (S) phosphorylation site, as described in *Arabidopsis* [12], *M. truncatula* [19], cucumber [21], and tea [37]. The first serine phosphorylation site is located between TMH1 and TMH2. However, one protein (MpSWEET2) contain threonine (T) and three proteins (MpSWEET16, -17, -19) contain aspartic (D) instead of serine (S). MpSWEET18 contain alanine (A). Except for alanine, these amino acids can all be phosphorylated. The second serine phosphorylation site is located between TMH5 and TMH6, and all MpSWEET proteins contain this site. In addition, the residues of the second MtN3/slv domain are more conserved than those of the first domain, the intramembrane region is highly conserved and the transmembrane region is relatively conserved (Figure 2), as described in soybean [25].

### 2.3. Conserved Motif and Gene Structure Analyses of MpSWEET Genes

The MEME server identified 15 conserved motifs in MpSWEET proteins. As shown in Figure 3A, most members of MpSWEET proteins contain motifs 1, 2, 3, 4, 5, 6, and 7, indicating that these motifs are highly conserved structures in the MpSWEET transporters. Except for MpSWEET01, MpSWEET10, MpSWEET16, and MpSWEET17, the MpSWEET members contain motifs 5, indicating that these gene family members may lose this motif in the process of differentiation from their common ancestor. Generally, MpSWEET proteins with similar motifs composition tend to cluster together. Motif 12 and motif 14 exist only in Clade I, motif 9 exists only in Clade II, and motif 13 and motif 15 exists only in Clade III. In addition to the seven highly conserved motifs, MpSWEET21 (Clade IV) also contains one specific motif 11. Motif 8 and motif 10 only appear in Clade I and III. It can be seen that the four Clades members of the evolutionary tree contain their specific motifs, indicating that our classification is reasonable.

The gene structure of all *MpSWEET* genes was investigated for the intron phases and exon-intron organization. Gene structural diversity plays a key role in the evolution of the SWEET gene family [42]. The pattern of exon-intron classification is consistent with the phylogenetic tree (Figure 3B). The majority of *MpSWEET* genes (13/23) contained five introns. The exon lengths are similar, while the intron lengths vary, with five *MpSWEET* genes (MpSWEET01/04/13/18/21) containing very long introns. The number of exons is between 4~6, of which 6 (accounting for 69%) are the majority. *MpSWEET10*, *MpSWEET16*, and *MpSWEET17* contain the fewest introns and exons (Figure 3B).

### 2.4. Chromosomal Distribution, Collinearity, and Ka/Ks Calculation

To investigate the relationship between gene duplication and genetic divergence within the *MpSWEET* genes, we determined the chromosomal locations of *MpSWEET* genes. The results showed that most *MpSWEET* genes were located proximal or distal to chromosomes, with Chr2 having the highest number of *MpSWEET*, with nine *MpSWEET* genes; Chr1 and Chr7 had the lowest number, with only one *MpSWEET* gene (Figure 4). The synteny regions on all seven chromosomes were analyzed to reveal the *MpSWEET* gene duplication events. Four pairs of genes (*MpSWEET01*/*18*, *MpSWEET02*/*12*, *MpSWEET09*/*15*, and *MpSWEET10*/*14*) were found to be segmentally duplicated, and three pairs of genes (*MpSWEET05*/*06*, *MpSWEET10*/*11*, and *MpSWEET18*/*19*) were assigned to tandem duplication.

A collinear map of *SWEET* genes in *M. polymorpha*, *Arabidopsis*, *M. truncatula*, and rice was constructed to further infer the phylogenetic mechanism of *MpSWEET* gene members. The red line indicated the orthologous *SWEET* gene between *M. polymorpha* and *Arabidopsis*, *M. truncatula*, and rice (Figure 5). The comparison between collinear orthologs and all orthologs can reveal how gene orders are conserved. A total of sixteen *MpSWEET* genes exhibit the highest level of collinearity relationship with *MtSWEET* genes. Except for chromosome 5 of *M. truncatula*, all chromosomes of *M. truncatula* have the *MpSWEET* homology genes (Figure 5, Appendix A). Thirteen *MpSWEET* genes exhibit a collinear relationship with the *AtSWEET* genes, and there is no gene on chromosome 7 of *Arabidopsis*, that are orthologous *MpSWEET* genes. Only four *MpSWEET* genes exhibit a collinear relationship with the *OsSWEET* genes.

The nonsynonymous/synonymous substitution ratio (*Ka*/*Ks*) of seven duplicated *MpSWEET* gene pairs between *M. polymorpha* was calculated. As shown in Table 2, the *Ka*/*Ks* ratios are less than 0.5 for all *MpSWEET* gene pairs, suggesting these genes have mainly undergone purifying selection during the plant’s evolution [43]. It took between 21.85 and 78.61 dates (MYA) for these genes to segmental duplication, which indicates that the main reason for the expansion is the main fragment duplication event and gradually shifted to tandem replication during the evolution of *M. polymorpha*. In addition, the evolution of the *SWEET* gene family in the *M. polymorpha* genome is highly conserved. Among them, *MpSWEET05*/*06* is the first tandem-replicated gene pair (Table 2).

### 2.5. Analysis of Promoter Cis-Acting Elements of MpSWEET Gene

To investigate the potential functions of *MpSWEET* genes, the sequences of *cis*-regulating elements related to environmental stress, plant hormones, and plant tissue growth responses in the promoter sequence (2000 bp) of each *MpSWEET* gene were obtained and analyzed. Six types of phytohormone-responsive *cis*-elements were identified in the promoter regions of *MpSWEET*, including two auxin-responsive elements, two gibberellin-responsive elements, one abscisic acid-responsive element, one salicylic acid-responsive element, and one MeJA-responsive element (Figure 6). Stress-responsive *cis*-elements were determined, including low temperature, drought, and wounding. Tissue-specific expression elements includ seed-specific regulation elements, meristem expression, endosperm expression, and palisade mesophyll cells. Light response elements includ light responsiveness and circadian control elements. As shown in Figure 6, all *MpSWEET* genes possess seven types of *cis*-elements related to hormone response. Wound-responsive elements are only distributed in clade III, including *MpSWEET07*, *MpSWEET12*, and *MpSWEET23*. Flavonoid biosynthetic genes regulation elements were only identified in *MpSWEET07*. For tissue-specific expression elements, endosperm and meristem-specific expression elements are distributed in clades I, II, and III; seed-specific regulation elements and Palisade mesophyll cells were only identified in *MpSWEET12* and *MpSWEET20* of clade III, respectively. Members of clade II do not contain the circadian control element; only the light responsiveness element was identified.

### 2.6. Protein–Protein Interaction Network

The MpSWEET proteins interaction network with unknown functions was constructed against the background of the model plant *Arabidopsis*, and the mutual regulation of proteins was studied by identifying highly correlated proteins, which was helpful to further explore the biological function and molecular regulatory network of *MpSWEET* genes, as studied in Triticum aestivum [44,45]. The protein network interaction showed that there were eleven directly related functional proteins and five indirect interacting proteins between MpSWEETs and AtSWEETs, which were SWEET1, SWEET2, SWEET3, SWEET4, AtVEX1, SWEET7, SWEET9, SWEET12, SWEET13, SAG29, SWEET17, and SWEET11, PTEN1, cwINV4, SUC2, AT1G23300. As shown in Figure 7, it was found that a total of 13 were sugar transporter proteins, except for AT1G23300, cwINV4, and PTEN1. Of these 13 MpSWEET proteins, there is no ortholog protein with SWEET11. Each protein node has an interactive relationship with the other. SWEET11 and SWEET17 are located at the center of the protein interaction network, and there were interactions with nine proteins, respectively, of which SWEET1, SWEET3, SWEET9, SWEET12, and SWEET13 are common to these two proteins. Moreover, seven MpSWEET proteins, SAG29, SWEET1, SWEET4, SWEET9, SWEET11, SWEET12, and SWEET13, show interactions with SUC2 (a phloem-specific plasma membrane sucrose transporter), suggesting that they are likely to cooperate and/or complete one or some physiological processes such as phloem loading of sucrose in leaves [46].

SWEET17 exports fructose out of the vacuole and is a major factor controlling fructose content in *Arabidopsis* leaves and roots [36]. MpSWEET21 was the homology of SWEET17 of *Arabidopsis*, predicting that MpSWEET21 may have a similar function. Studies have shown that AtSWEET11 and AtSWEET12 are also involved in cold stress or water deficit conditions. The *AtSWEET11*/*12* double mutants exhibited greater freezing tolerance than the wild-type and both single mutants [47]. There are four proteins (MpSWEET15, MpSWEET07, MpSWEET12, and MpSWEET03) that are orthologs of SWEET proteins *Arabidopsis* of (SWEET1, WEET9, SWEET12, and SWEET13), suggesting that these proteins may be involved in plant growth and development and stress response through similar regulatory mechanisms.

### 2.7. In Silico Analysis of the Expression Patterns of MpSWEET

The expression profiles of the *MpSWEET* genes in different developmental stages/tissues and stress treatments were analyzed based on the microarray data from *M. truncatula* and *M. sativa* and RNA-Seq data of *M. polymorpha*. Cluster analysis of expression data showed that *MpSWEET* genes had different transcript levels in different tissues/stages, exhibiting tissue-specific expression patterns that were similar to SWEET gene expression patterns in pear [48]. As shown in Figure 8A, five *MpSWEET* genes were expressed in at least one of the tested tissues (including flower, leaf, petiole, stem, and pod). *MpSWEET02*, *MpSWEET03*, and *MpSWEET12* exhibited relatively higher expression in flower and leaf; *MpSWEET21* exhibited relatively higher expression in leaf, stem, and petiole. Several *MpSWEET* genes displayed obviously high expression levels in only one tissue. For example, *MpSWEET04*, *MpSWEET07*, and *MpSWEET15* were preferentially expressed in flowers (Figure 8A). Among them, *MpSWEET15* had the strongest flower–tissue specificity, with an expression level in the flower six times higher than that in the leaf. *MpSWEET23* exhibited much higher transcript abundance in the large pod than in other tissues, which was 14~146 times higher than that in flower and leaf, and the difference in expression level in different tissues may be related to the execution of a certain step in the process of sugars transport [15] and specific function to tissue development [11].

*MpSWEET05* was only highly expressed in the early flowering stage of *M. polymorpha*. *MpSWEET03* exhibited highly expressed in all three stages, and *MpSWEET02* and *MpSWEET21* were gradually highly expressed in the early flowering to the late flowering stage (Figure 8B). Eleven *MpSWEET* genes (*MpSWEET01*, *MpSWEET04*, *MpSWEET06*, *MpSWEET08*, *MpSWEET11*, *MpSWEET13*, *MpSWEET15*, *MpSWEET16*, *MpSWEET17*, *MpSWEET20*, *MpSWEET23*) did not have corresponding probe sets in the dataset. The expression levels of *MpSWEET19* and *MpSWEET21* under salt stress were significantly higher than those of the control, and *MpSWEET07* was also upregulated. *MpSWEET12* was upregulated under drought treatment (Appendix A).

### 2.8. Validation of MpSWEET Gene Expression Patterns Using qRT-PCR

Combining the results of protein interaction networks and expression patterns, six genes (*MpSWEET03*, *MpSWEET05*, *MpSWEET07*, *MpSWEET12*, *MpSWEET15*, and *MpSWEET21*) were selected to examine expression profiles via qRT-PCR in leaf under drought, salt, and cold stresses. As shown in Figure 9, the expression of the selected *MpSWEET* genes was significantly changed by the three stress treatments. Among the six *MpSWEET* genes, *MpSWEET03* showed significantly lower expression levels under drought stresses (24 h) and cold stresses than the control. *MpSWEET05* showed similar expression patterns under drought and cold treatments and peaked at 24 h. *MpSWEET07* and *MpSWEET12* showed similar expression levels under salt and cold stress treatments, respectively. *MpSWEET07* was expressed the highest at 12 h salt stress, with an expression level about 4000 times that in control. The expression of *MpSWEET15* and *MpSWEET21* were significantly upregulated under three stress treatments (Figure 9), consistent with RNA-Seq data (Appendix A). *MpSWEET21* showed similar expression levels, with an expression level at 12 h drought stress nearly 70 higher than that in control.

The results showed that, in leaves, the expression of *MpSWEET05*, *MpSWEET07*, *MpSWEET12*, *MpSWEET15*, and *MpSWEET21* was significantly upregulated at cold and salt stress compared to that of the control, particularly that of *MpSWEET07*, *MpSWEET12*, and *MpSWEET21*, whose expression peaked at 12 or 24 h (Figure 9). *MpSWEET07* possessed opposite expression profiles in drought conditions compared to cold and salt treatments. The expression level of *MpSWEET05*, *MpSWEET12*, *MpSWEET15*, and *MpSWEET21* increased to varying degrees under drought stress. These results indicated that these genes might participate in abiotic stress response by utilizing different regulatory mechanisms to modulate sugar levels.

## 3. Discussion

SWEET proteins are widely distributed in the plant kingdom and regulate diverse physiological and biochemical processes, particularly source-sink interactions [3,49]. Typically, angiosperm genomes contain about 15~25 *SWEET* genes [14]. The previously reported number of *SWEET* genes was 17 in *Arabidopsis* [10], 21 in rice [10,17], and 25 in *M. truncatula* [19]. In this study, a total of 23 *SWEET* genes were identified from *M. polymorpha*, which were more similar to the number of *MtSWEET* genes because the isolation time of *M. polymorpha* and *M. truncatula* is relatively close (nearly 15.3 million years ago) [39]. We found that MpSWEET and MtSWEET proteins were highly similar and arranged next to each other in the phylogenetic tree (Figure 1). Similar to collinearity analysis results, they showed strong genomic syntenic relationships between *M. polymorpha* and *M. truncatula*. Previous studies showed that chromosome 3 of *M. polymorpha* arose from the fusion of chromosomes 3 and 7 of *M. truncatula* [39]. In this study, the number of *MpSWEET* genes in chromosome 3 is exactly the sum of chromosomes 3 and 7 of *M. truncatula* [19], indicating that the *SWEET* gene family is highly conserved in the evolution of the genus *Medicago*.

Similar to other higher plants [10,19,25], MpSWEETs can also be classified into four clades. Clade III was the largest group, including nine members, and clade IV only contained one member (Figure 3). It has been proposed that gene duplications, including segmental and tandem duplications, play crucial roles in the evolution of various organisms [50]. In this study, seven pairs of *MpSWEET* genes were involved in the duplications event, including tandem and segmental duplications. The *Ka*/*Ks* ratio of all *MpSWEET* gene pairs involved in duplication was less than 1, suggesting *MpSWEET* genes had primarily experienced purifying selection in their evolutionary histories, with slight variation after duplication [13]. Most *MpSWEET* genes possessed six exons and five introns (Figure 3B), and most *MpSWEET* genes in the same subclass had similar structures, which is consistent with the results in other plants, including litchi [30], cabbage [51], and tomato [20], suggesting that the *SWEET* genes have been highly conserved during evolution. Furthermore, two possible serine phosphorylation sites were found on the inner sides of the membrane area of MpSWEET proteins (Figure 2), indicating that the inner sides of the membrane area of the MpSWEET proteins are likely to be their important functional areas or their active regulatory regions [21]. Additionally, each clade had more similar motif compositions at the conserved N-terminal, and most MpSWEETs had seven highly conserved motifs (motif 1~motif 7) (Figure 3A), but none in the diversified C-terminal ends. We also found that many putative phosphorylation sites were identified in the C-terminal cytosolic region of *M. polymorpha*, which speculates that these phosphorylation sites may play an important role in the regulation of MpSWEET activity.

The protein interaction network analysis results showed a complex regulatory network between directly functional proteins, which regulated plant growth and development through the transport of sucrose and other substances (Figure 7). *MpSWEET03* and *MpSWEET12* were highly expressed in leaves (Figure 8A), and it is predicted that they may be related to phloem loading and the long-distance transport of sucrose in *M. polymorpha*, just like their orthologs such as *AtSWEET12* [15] and *MtSWEET12* [19]. Similarly, the interaction analyses of MpSWEETs demonstrated that MpSWEET03 was directly related to SUC2, forecasting that it may play an important role in phloem loading of sucrose [46]. It is reported that the *SWEET* family is involved in the fruit development and seed-ripening process of apples [52], litchi [30], and other plants [29]. Previous studies found soybean *GmSWEET15* is specifically expressed in the endosperm at the cotyledon stage, mediating the transport of sucrose from endosperm to embryo, and *GmSWEET15* mutation will inhibit endosperm degradation and embryo growth and development and also lead to soybean seed abortion [53]. *SWEET11*/*12*/*15* of *Arabidopsis* exhibit seed-specific expression patterns, with the function to seed filling [28]. MpSWEET12 is highly homologous to soybean GmSWEET15 and SWEET12 (Figure 7), which speculates that its function may be related to the sugars transport of the embryo to support seed growth and development of *M. polymorpha*. Seed-specific regulation elements were only identified in *MpSWEET12* (Figure 6), supporting that conjecture. Additionally, the expression level of *MpSWEET03* in the flower was nearly 5-fold higher than that in the root (Figure 8A), and it was highly expressed throughout the growth stage from the seedling to the late flowering stage (Figure 8B). *AtSWEET13* and *AtSWEET14* are necessary for the normal development of anthers, seeds, and seedlings [54], MpSWEET03 is an ortholog of AtSWEET13, indicating that it may play a key role in sugar transport the vegetative growth stage as well as the reproductive growth stage of *M. polymorpha*. The expression of the *MpSWEET07* gene in the flower is seven times that of the leaf and 291 times that of the root (Figure 8A). AtSWEET9 is a sucrose efflux transporter and was shown to function in the secretion of sucrose for nectar production [11]. MpSWEET07 is an ortholog of AtSWEET9, and it can be speculated that *MpSWEET07* may have a similar function to *AtSWEET9*. The expression level of *MpSWEET15* in the flower was nearly five times more than that in the root. MpSWEET15 is an ortholog of AtSWEET1, *AtSWEET1* was highly expressed in flower, and expression in roots was low, indicating that protein may supply nutrients to the gametophyte or nectary [10]. The result of protein–protein interaction network showed that three proteins were directly related to each other, suggesting that they may play an important in sugar transport processes together in the reproductive growth stage of *M. polymorpha*.

The transport of sugars from the source to the library can alter carbohydrate distribution and homeostasis, contributing to plant tolerance to abiotic stresses [55]. A growing body of evidence suggests that SWEETs are widely involved in abiotic stress responses, such as wheat [56], banana [57], and *Poa Pratensis* [58]. In the present study, high-expression genes at different developmental stages and tissue-specific expression *MpSWEET* genes exhibited extensive responses to cold, salt, and drought stresses (Figure 9), and similar results were also observed in *M. truncatula* [19], banana [57], and tea [37]. It is worth noting that the expression levels of four genes (*MpSWEET05*, *MpSWEET12*, *MpSWEET15*, and *MpSWEET21*) were significantly altered by three stress treatments, and MpSWEET07 was significantly increased at cold and salt treatments, being consistent with RNA-Seq data (Appendix A). AtSWEET15 was found to be associated with cell viability under high salinity and other osmotic stress conditions [35]. The results in this paper are similar to this observation. As its homolog, *MpSWEET05* was highly expressed at 12 h salt stress (Figure 9). As the fructose-specific transporter, AtSWEET17 plays a primary role in fructose homeostasis following 1 week of 4 °C treatment [36]. MpSWEET21 is a highly ortholog of AtSWEET17, and *MpSWEET21* was upregulated by these three stress treatments, implying that MpSWEETs have evolved different mechanisms to adapt to various abiotic stresses in legumes. All these results indicated a positive relationship between the protein network interaction prediction and qRT-PCR during the analysis of the potential functions of MpSWEETs under three abiotic stresses in M. polymorpha. Taken together, MpSWEET proteins not only participate in the growth and development of *M. polymorpha* but also may participate in abiotic stress response by regulating other MpSWEETs. Based on the results of the above analysis, we speculate that MpSWEET21 may be the most important core protein in *M. polymorpha*, which plays a vital role in regulating fructose content in plant leaves and sugar transport in flower organs and regulating plants growth under stress. Moreover, there were four highly expressed genes from subclass III, indicating that this subclass may play an important role in regulating *M. polymorpha* growth and abiotic stress response, and it is necessary to further analyze the functional SWEET protein in the genomic of *M. polymorpha* to improve the application and promotion of *M. polymorpha* as a forage crop and vegetable.

## 4. Materials and Methods

### 4.1. Database Mining and Identification of SWEET Family Genes in M. polymorpha

The *M. polymorpha* whole-genome sequence was downloaded from the NGDC database (https://ngdc.cncb.ac.cn/, accessed on 11 October 2021). SWEET protein sequences of 17 AtSWEETs, 25 MtSWEETs, and 21 OsSWEETs were obtained from the TAIR (http://www.arabidopsis.org/, accessed on 13 October 2021), Phytozome 13 (https://phytozome-next.jgi.doe.gov/, accessed on 20 October 2021), and RGAP (http://rice.plantbiology.msu.edu/, accessed on 22 October 2021) databases, respectively, and used as a query to search against the *M. polymorpha* proteome by the BLAST. The Hidden Markov Model (HMM) corresponding to the MtN3/saliva domain (PF03083) was retrieved from the InterPro website (https://www.ebi.ac.uk/interpro/, accessed on 12 May 2022), as described in the identification of *TaTALE* genes [59]. The redundancy protein sequences of SWEET members were removed from the Expasy website (https://web.expasy.org/decrease_redundancy, accessed on 15 May 2022) and the proteins with very short amino acid sequences (<150 aa) were excluded. Additionally, the SMART database (https://smart.embl.de/, accessed on 12 June 2022) was then used to further filter and analyze the non-redundant SWEET protein sequences to validate the HMM and BLAST search results [60].

### 4.2. In Silico Sequence Analysis

The molecular weights (MWs), grand average hydropathicity (GRAVY), and isoelectric points (pIs) of the MpSWEET proteins were detected using SMS2 online software (http://www.detaibio.com/, accessed on 8 July 2022). The transmembrane helices were predicated on the TMHMM server2.0 using a method based on a hidden Markov model (https://services.healthtech.dtu.dk/, accessed on 20 September 2022). The details of TMHs are listed in Appendix A. The subcellular localization was determined by WoLFPSORT (https://www.genscript.com/, accessed on 28 September 2022). The protein sequences of 23 MpSWEETs, 17 AtSWEETs, 25 MtSWEETs, and 20 OsSWEETs were aligned through the ClustalW program using the format parameters at MEGA 7.0. A phylogenetic tree was then constructed using the neighbor-joining (NJ) method with 1000 bootstrap replicates [61]. The *Ka*/*Ks* ratio calculated by TBtools [62] was used to show the selection pressure for the duplicate genes, and the formula is *Ks*/2 × 1.5 × 10^−8^ [43]. The promoter regions (2000 bp upstream of the translation initiation codon) of *MpSWEET* genes were obtained from the *M. polymorpha* genome data, and the *cis*-acting elements were identified using the PlantCARE database (http://bioinformatics.psb.ugent.be/, accessed on 12 June 2022).

### 4.3. Multiple Sequence Alignment, Conserved Motif Prediction, and Protein–Protein Network Interaction

Multiple sequence alignments of the amino acid sequences of MpSWEET proteins were generated using the DANMANversion 9 program. The MEME online program version 5.4.1 (http://meme-suite.org/, accessed on 13 September 2022) was used to determine the conserved motifs of *M. polymorpha* SWEET proteins, and the maximum number of motifs was set to 15. Ultimately, the motifs were visualized using TBtools [62]. *Arabidopsis* was selected as the background to build protein–protein interaction network structure by STRING online software (https://cn.string-db.org/, accessed on 1 November 2022). The meaning of the network edges was set as a line color to indicate the type of interaction evidence. The minimum required interaction score was 0.400 with medium confidence, and the maximum number of interactors was 10. Similar functions of *MpSWEET* genes were analyzed using string protein interaction network structure. Furthermore, protein–chemical interaction was predicted by using STITCH (http://stitch.embl.de/, accessed on 25 April 2022), the maximum number of interactors was set to 10, and the minimum required interaction score is 0.400. However, no interaction of SWEET proteins with chemicals was found.

### 4.4. Analysis of Gene Structure, Chromosomal Distribution, and Collinearity

A schematic diagram of the gene structure of *MpSWEET* genes was generated by Gene Structure Display Server (GSDS, http://gsds.gao-lab.org/, accessed on 10 November 2022) using the coding sequence (CDS) with their corresponding genomic DNA (gDNA) sequences file information. The chromosome location data of *MpSWEETs* was retrieved from GFF3 of *M. polymorpha* and was mapped with TBtools software. The genome sequences of *M. polymorpha* were compared with those of *Arabidopsis*, *M. truncatula*, and rice, respectively, and combined with the whole genome chromosome position information of these three species. The *MpSWEET* family genes chromosome distribution and interspecific collinearity relationship were obtained using MCScanX. Details of collinearity are listed in Appendix A.

### 4.5. Expressed Sequence Tag Retrieval in Medicago Plants

To predict the tissue expression profile of the *MpSWEET* genes, genome-wide microarray data from *M. truncatula* in different tissues were retrieved from *M. truncatula* Gene Expression Atlas (https://www.zhaolab.org/LegumeIP/gdp/13/gene, accessed on 6 December 2022), and then, using the BLAST tool to match the orthologous gene of *MpSWEET* family genes, the transcript data of the *MpSWEET* genes were analyzed. The RNA-Seq data of the aboveground parts of *M. polymorpha* at three different growth stages (including seedling stage, early flowering stage, and late flowering stage) were downloaded to analyze the developmental expression profile of the *MpSWEET* genes.

Additionally, to analyze the potential function of *MpSWEET* in response to stress, Alfalfa Gene Editing Database (http://alfalfagedb.liu-lab.com/heatmap/heatmap/, accessed on 1 December 2022) was used to predict the expression levels of 12 orthologous *MpSWEET* genes of *M. sativa* under salt and drought stress treatments. The normalized expression data were used to generate a heatmap using the TBtools software [62]. The expression profile is listed in Appendix A. The information on orthologous *MpSWEET* genes in *M. truncatula* and *M. sativa* are listed in Appendix A.

### 4.6. Plant Growth, Treatments, RNA Isolation, and qRT-PCR Analysis

Seeds of *M. polymorpha* were obtained from Yangzhou University. The seeds were germinated on a wet germinated disc for 3 days at 4 °C and were then transferred to a liquid culture medium. For cold stress treatment, four-week-old *M. polymorpha* seedlings were maintained at a constant temperature of 4 °C. For drought and salt treatments, the seedlings were irrigated with 20% PEG 6000 and 220 mM NaCl. Three biological replicates were conducted for each treatment, and each replication contained 10 plants. The leaves were collected at 0, 3, 12, and 24 h after the three stress treatments, respectively, and were immediately frozen in liquid nitrogen for RNA extraction.

For a more in-depth look at the *MpSWEET* gene expression patterns under abiotic stress, total RNA was extracted from the leaf tissues using the RNA prep Pure Plant. The total RNA Extraction Kit (Tiangen, Beijing, China) was used according to the manufacturer’s instructions. The total RNA was synthesized into the first strand of cDNA using the cDNA synthesis kit (Vazyme, Nanjing, China). Six pairs of gene-specific primers for qRT-PCR analysis were designed by PerlPrimer v1.1.21 software and displayed in Appendix A. A 10 μL reaction volume for each sample containing 5 μL of 2 × AceQ Universal SYBR qPCR Master Mix (Vazyme, Nanjing, China), 0.2 μL of each primer, 1 μL of diluted cDNA product, and 3.6 μL of ddH_2_O. The qRT-PCR reaction procedures carried out on the platform supported by QuantStudio 3 system (Thermo Fisher Scientific, Waltham, MA, USA) were as follows: 5 min at 95 °C for DNA Polymerase activation, denaturation, and anneal/extension at 95 °C for 10 s and 60 °C for 30 s, respectively, for a total of 40 cycles.

The *M. polymorpha ACTIN* gene (*Mpo3G42410*) was selected as the internal control to calculate the relative expression data according to the 2^−∆∆CT^ method [63]. One-way Analysis of Variance (ANOVA) tests for qRT-PCR data were performed using IBM^®^ SPSS^®^ Statistics 25 (IBM, Armonk, NY, USA) software. The data were generated in three biological replicates from the leaves tissue, and standard errors of means among replicates were also calculated. The plots were produced by GraphPad Prism 8 (GraphPad Software, Boston, MA, USA).

## 5. Conclusions

In this study, 23 *MpSWEET* genes were identified in the *M. polymorpha* genome and further phylogenetically clustered into four clades. The results showed that the *SWEET* genes of *M. polymorpha* were highly conserved in the evolution of annual alfalfa. *MpSWEET05*, *MpSWEET07*, *MpSWEET12*, *MpSWEET15*, and *MpSWEET21* were involved in various physiological processes of *M. polymorpha*, especially in regulating reproductive development, including flower and seed development, and exhibited high expression levels in at least two of the three abiotic stress treatments. Although the exact role of most of the *MpSWEETs* identified in this study has not yet been established, it is conceivable that genes could modulate sugar levels by particular mechanisms and thus may orchestrate stress tolerance in plants. Overall, these findings facilitate unraveling the potential candidate *MpSWEET* genes involved in the response to abiotic stress and provide important clues for further studying the biological functions of the MpSWEET proteins of *Medicago* plants in the future.

## Figures and Tables

**Figure 1 plants-12-01948-f001:**
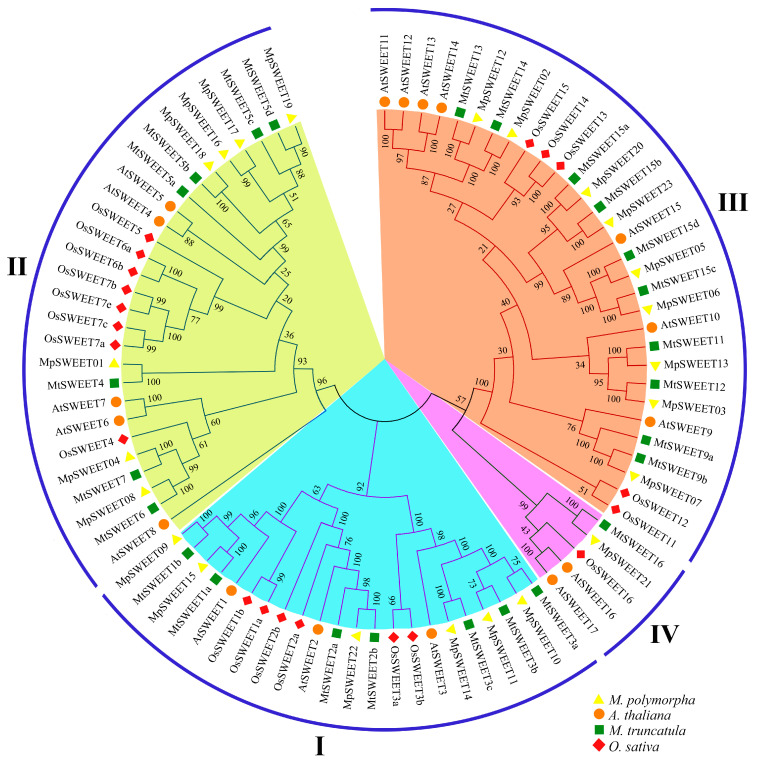
Phylogenetic tree of the *SWEET* family from *M. polymorpha*, *M. truncatula*, *Arabidopsis thaliana*, and *Oryza sative*. The yellow triangle, orange dot, green rectangles, and red rhombuses color represent *M. polymorpha* and *A. thaliana*, *M. truncatula*, and *O. sative*, respectively.

**Figure 2 plants-12-01948-f002:**
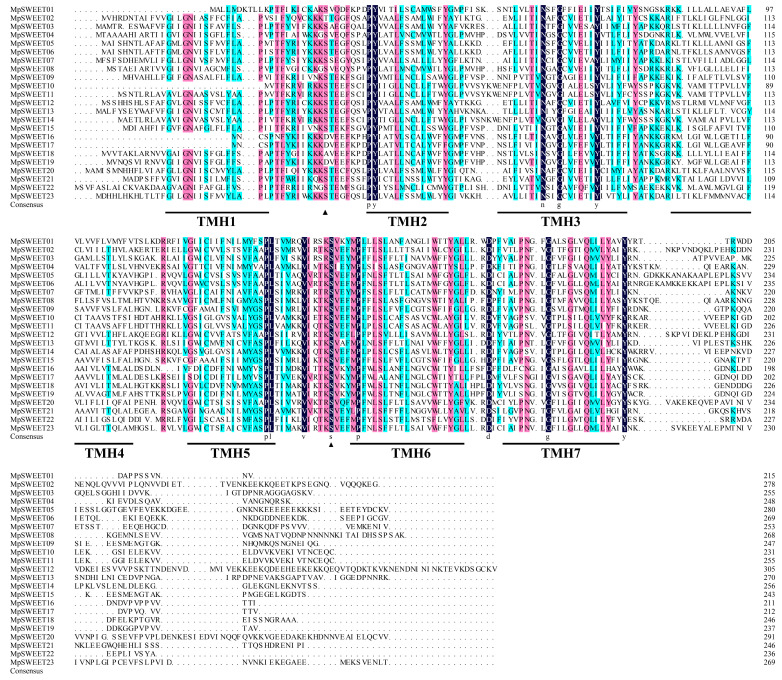
Amino acid sequence alignment results of 23 conserved domains of SWEETs in *M. polymorpha*. The positions of the seven transmembrane domains (TMH1 to TMH7) are represented by black line segments. The serine phosphorylation sites are represented by black triangles. The residues indicated in black were fully conserved among all proteins.

**Figure 3 plants-12-01948-f003:**
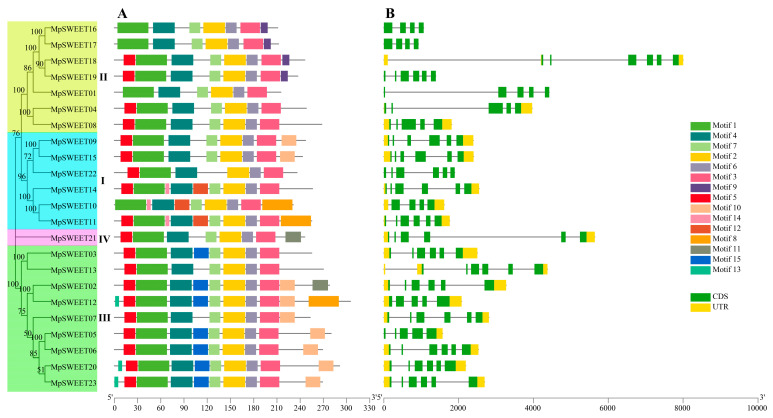
Conserved motifs (**A**), and gene exon-intron structures (**B**) of *SWEET* family members in *M. polymorpha*.

**Figure 4 plants-12-01948-f004:**
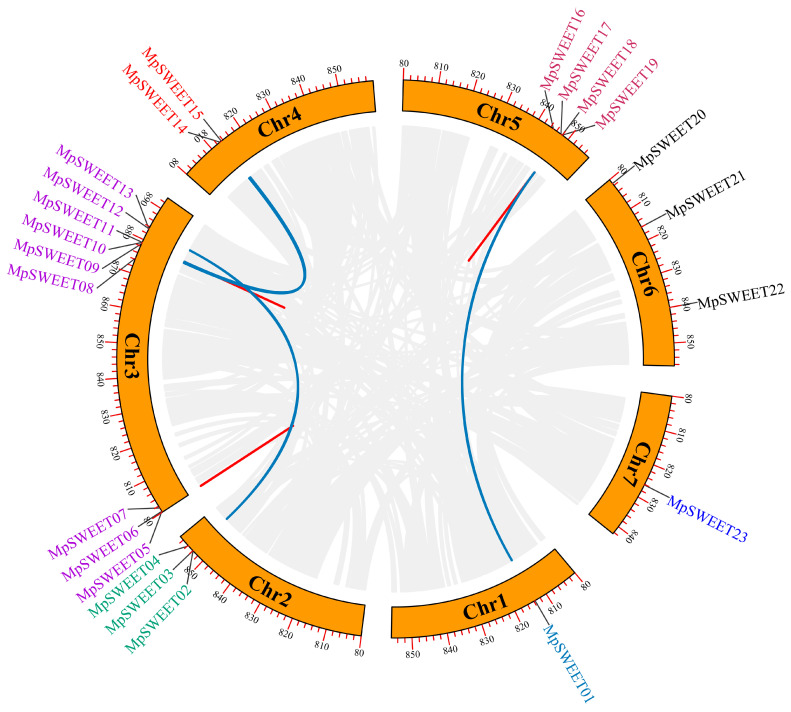
Chromosome distribution and replication event of *SWEET* genes in *M. polymorpha*. The segmentally duplicated genes are connected by blue lines, and tandem duplicated genes are connected by red lines.

**Figure 5 plants-12-01948-f005:**
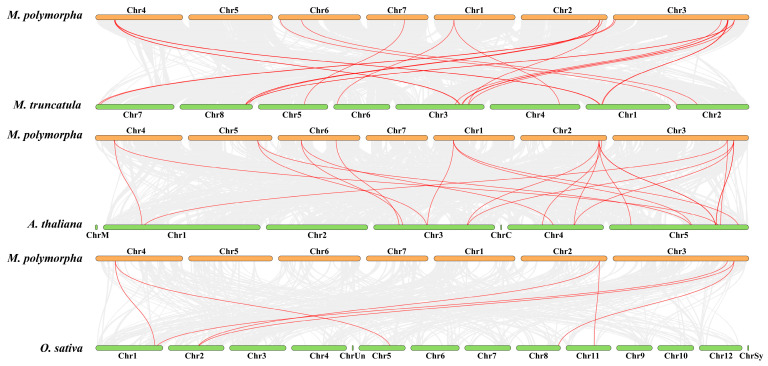
Collinearity analysis between *M. polymorpha* and *A. thaliana*, *M. truncatula*, and *O. sative*.

**Figure 6 plants-12-01948-f006:**
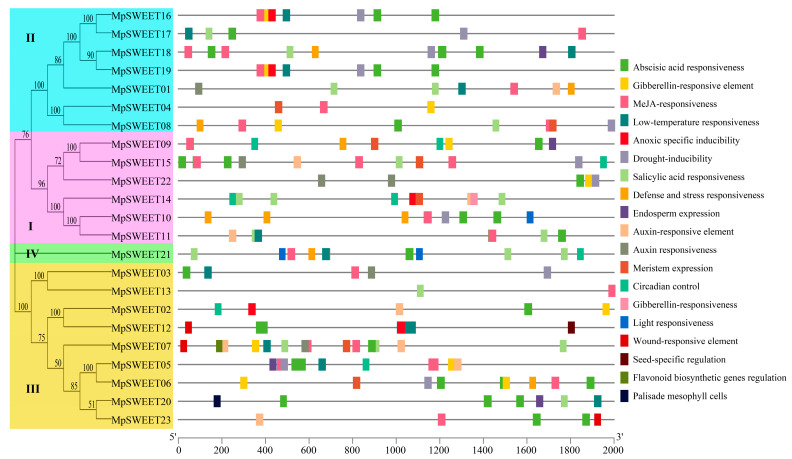
The *cis*-acting elements of the promoter region form *SWEET* genes in *M. polymorpha*.

**Figure 7 plants-12-01948-f007:**
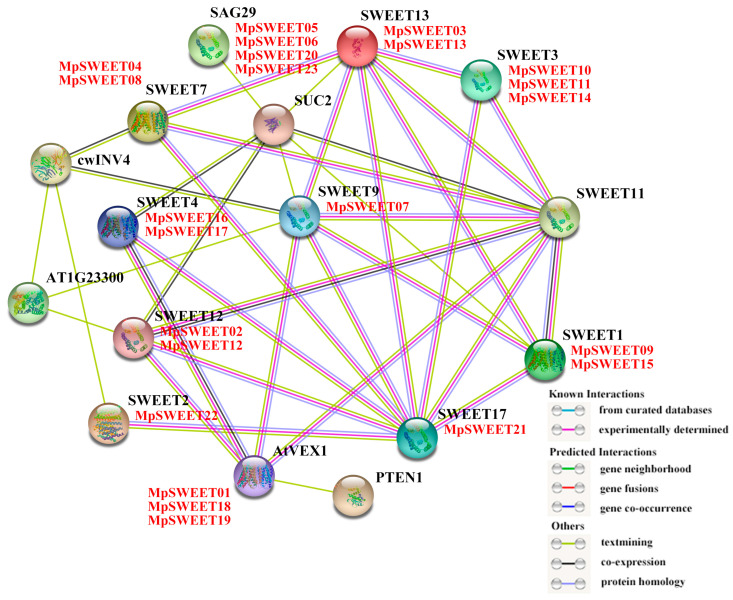
Protein interaction networks of 23 MpSWEET proteins based on orthologs in *A. thaliana*. Red font means orthologous MpSWEET proteins in *A. thaliana*.

**Figure 8 plants-12-01948-f008:**
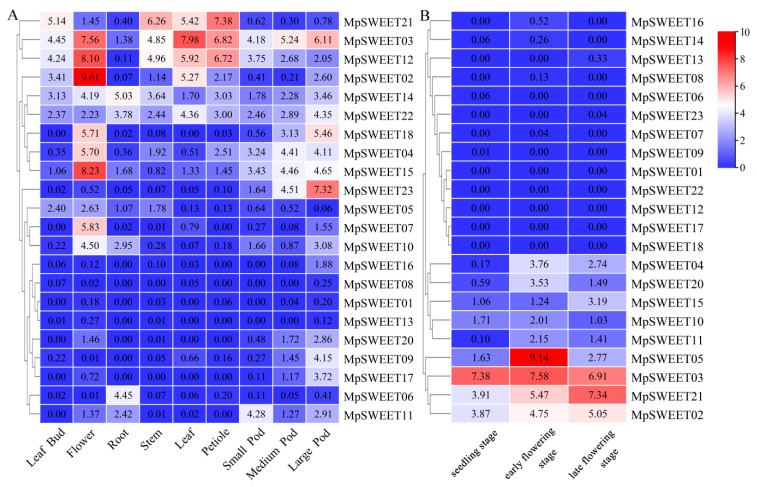
Expression patterns and hierarchical clustering of *MpSWEET* genes. (**A**) The expression levels of the orthologs in *M. truncatula of MpSWEET* genes in different tissues; (**B**) the expression profiles of *MpSWEET* genes at three developmental stages.

**Figure 9 plants-12-01948-f009:**
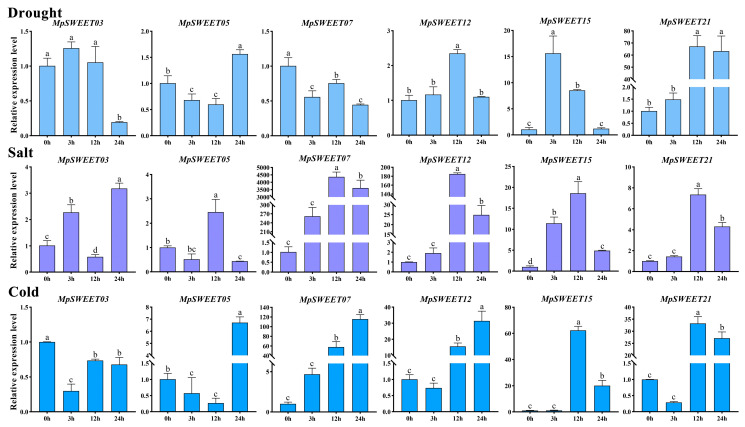
Expression of *MpSWEET* genes in response to three stresses of drought, salt, and cold. The *M. polymorpha ACTIN* gene (*Mpo3G42410*) was used as a standard control, and the 2^−∆∆CT^ method was used to calculate the relative levels of gene expression. Data were statistically analyzed using Duncan’s test with SPSS25 and different letters indicate statistically significant differences (*p* < 0.05).

**Table 1 plants-12-01948-t001:** Physical and chemical characteristics of SWEET transporter members in *M. polymorpha*.

Gene Name	Gene ID	Chromosomal Location	CDS Size (bp)	Protein Length (aa)	Molecular Weight (kDa)	Protein GRAVY	pI	Aliphatic Index	Subcellular Localization	TMHs
MpSWEET01	Mpo1G11620	chr1	648	215	24.51	0.726	10.04	122.37	cyto_nucl	6
MpSWEET02	Mpo2G13510	chr2	837	278	31.33	0.448	8.45	116.69	chlo	7
MpSWEET03	Mpo2G13500	chr2	768	255	28.53	0.886	10.21	123.1	chlo	7
MpSWEET04	Mpo2G11620	chr2	747	248	27.24	0.771	9.62	123.75	plas	6
MpSWEET05	Mpo3G0660	chr3	843	280	31.4	0.446	8.42	111.75	chlo	7
MpSWEET06	Mpo3G0670	chr3	810	269	30.19	0.491	9.23	115.95	chlo	7
MpSWEET07	Mpo3G1330	chr3	762	253	28.59	0.789	8.17	121.3	vacu	7
MpSWEET08	Mpo3G49060	chr3	807	268	29.39	0.599	9.31	111.23	plas	7
MpSWEET09	Mpo3G45160	chr3	744	247	27.39	0.721	9.94	112.43	chlo	7
MpSWEET10	Mpo3G44840	chr3	696	231	25.6	0.645	8.81	121.73	plas	6
MpSWEET11	Mpo3G44830	chr3	768	255	28.03	0.707	8.86	124.47	chlo	7
MpSWEET12	Mpo3G40800	chr3	918	305	34.56	0.23	8.34	106.92	chlo	7
MpSWEET13	Mpo3G40790	chr3	813	270	30.38	0.662	9.32	118.78	chlo	7
MpSWEET14	Mpo4G5970	chr4	771	256	28.13	0.643	9.63	127.07	chlo	7
MpSWEET15	Mpo4G5790	chr4	732	243	26.74	0.805	9.6	111.11	chlo	7
MpSWEET16	Mpo5G25800	chr5	636	211	24.16	0.728	5.13	120.05	plas	5
MpSWEET17	Mpo5G1131L	chr5	639	212	23.96	0.77	6.47	124.01	plas	6
MpSWEET18	Mpo5G27400	chr5	741	246	27.3	0.695	9.54	118.82	plas	7
MpSWEET19	Mpo5G27410	chr5	714	237	26.54	0.659	9.02	112.57	plas	6
MpSWEET20	Mpo6G14700	chr6	876	291	32.88	0.72	7.33	124.16	plas	7
MpSWEET21	Mpo6G4600	chr6	741	246	27	0.588	8.86	119.63	plas	7
MpSWEET22	Mpo6G28390	chr6	711	236	26.37	0.894	8.28	121.44	vacu	7
MpSWEET23	Mpo7G11350	chr7	810	269	30.52	0.799	8.34	129.33	plas	7

**Table 2 plants-12-01948-t002:** Non-synonymous (*Ka*) and synonymous substitution rate (*Ks*) duplicated *SWEET* gene pairs in *M. polymorpha*.

Gene ID		*Ka*	*Ks*	*Ka*/*Ks*	Duplication Date (MYA)
MpSWEET01	MpSWEET18	0.4299	2.3583	0.1823	78.61
MpSWEET02	MpSWEET12	0.2788	0.6556	0.4254	21.85
MpSWEET09	MpSWEET15	0.1295	1.0653	0.1215	35.51
MpSWEET10	MpSWEET14	0.1639	0.9861	0.1662	32.87
MpSWEET05	MpSWEET06	0.1288	0.7699	0.1674	25.66
MpSWEET10	MpSWEET11	0.0281	0.2244	0.1253	7.48
MpSWEET18	MpSWEET19	0.1820	0.5691	0.3198	18.97

## Data Availability

Not applicable.

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
