# Peer review of "Genome-Wide Identification and Expression Analysis of the *SWEET* Gene Family in Annual Alfalfa (*Medicago polymorpha*)"

_plants, 2023, doi:10.3390/plants12101948_

Round 1
Reviewer 1 Report
The purpose of the article “Genome-wide identification and functional analysisofthe SWEET gene family in annual alfalfa (Medicago polymorpha)” is reported a detailed analysis of the M. polymorpha SWEET family members, including the phylogeny, conserved domain, gene structure, chromosome distribution, cis-acting regulatory elements, and expression patterns in different developmental stages/tissues as well as in response to drought, salt, and cold stresses.
Assessing your work, I am full of admiration for the enormity of work put into its implementation. The results are very interesting, which do not raise any objections of a scientific and substantive nature, they are very interesting and interesting from both a scientific and practical point of view. The work was prepared in a very careful way, it is obvious that you have mastered the scientific technique. Nonetheless, the article must be improved in terms of writing since some grammar and syntax errors are present in the manuscript.
My suggestions:
The report on M&M is very succinct! Provide experimental work plan at the start of M&M. No detail description is available about the experimental design.
What statistical analyzes are used?
Rewrite the conclusion! It needs to be much improved.
The article must be improved in terms of writing since some grammar and syntax errors are present in the manuscript.
Reviewer 2 Report
In this study authors have identified 23 SWEET transporter proteins in the Medicago polymorpha and used for in-silico characterization and expression analysis.
The Ms needs to be thoroughly revised before publication.
1. Ms is very casually written. What does it mean- RNA-Seq data show that someMpSWEET genes were specifically expressed in disparate developmental stages? Authors should specify the name of stages. Abstract needs to revise properly.
2. Full name should be first written and abbreviations should be in bracket.
3. Title needs to be changed. I could not see any functional data in the Ms
4. 2.1, Needs to elaborate how authors reached to number 23? It is not possible to get only this much hits during blast search. Is their any specific domain in this protein? Authors should confirm the domain in each identified transporter. Author may follow this Ms for domain confirmation https://www.mdpi.com/2223-7747/11/5/587
5. 2.2 Protein conserved domains analysis. I could not see any domain analysis here. Is it not the TMs analysis only? What is the roles of highlighted residues in MSA?
6. 2.3 Where the conserved motifs have been analysed? In gene or Proteins? Not clear from the results. Authors should write gene structure and protein chareacterization in separate para.
7. The red line indicated the homologous SWEET gene betweenM. polymorpha200 and Arabidopsis, M. truncatula, rice (Figure 5). These are homologous or orthologous?
8. 2.5 Analysis of promoter cis-acting elements of MpSWEET gene and protein network interaction. Make two different sections.
9. Interaction data seems to be very limited. Authors may perform co expression analysis (https://www.mdpi.com/1422-0067/23/23/14867) along with STRING.
10. What about the chemical interactions? That may be analysed by STITCH ?
11. Since the study is mostly in-silico, miRNA interaction can also be performed (https://www.mdpi.com/2075-1729/12/7/941)
Most of the scientific names are not in format. Authors should thoroughly check.
I could not find any statististical parameters explained in the Ms even in RT PCR data. What was the internal control? How many replicates were used.. etc.
Language needs to be improved significantly.
Reviewer 3 Report
The present article describes the identification, sequence analysis and transcriptional activity of SWEET genes in M. polymorpha. The work has a typical layout for this type of research. The authors conducted a systematic and comprehensive analysis of SWEET genes, which has significant reference value for follow-up in-depth molecular function research.
The authors undoubtedly put a lot of effort into research. The whole manuscript is well-written, straightforward and easy to follow. The materials and methods are sufficiently described. The results are correctly described and discussed as well.
However, I have a few comments:
- The names of plant species not in italics appear throughout the text (ex. lines 264, 386),
- Small punctuation errors appear in the text (ex. lines 131, 408)
- In chapter 2.6. the authors describe the expression results from the M. polymorpha RNA-Seq. In Fig. 8B, the description S1-S3 appears; but it doesn't mean what it means. Description of developmental stages appears only in materials and methods. Information about the stages of development should appear at this point in the manuscript.
- It is unclear whether the manuscript's authors perform the RNA-seq experiment or the expression date comes from some database like microarray results.
Round 2
Reviewer 2 Report
The Ms has been improved but a few sections needs further attention. The interaction data results should be properly explained in results. Authors may refer to www.mdpi.com/1422-0067/23/23/14867 ; and www.mdpi.com/2077-0472/13/4/783 , all the interacting parters should be elaborated in the results. The parameters can also be included in the method section. Further, the role of each interacting partners should be included in the discussion section. Authors may go through the suggested Ms for further clarity.
Moderate corrections are required.
